# Innovative Cutting and Valorization of Waste Fishing Trawl and Waste Fishing Rope Fibers in Cementitious Materials

**Ali Hussan** [1,*], **Badreddine El Haddaji** [1,2,3], **Mohammed Zelloufi** [1] **and Nassim Sebaibi** [1]

[1] COMUE NU, Builders Ecole d'Ingénieurs, Builders Lab, 14610 Epron, France; elhaddaji@esitc-paris.fr (B.E.H.); mohammed.zelloufi@builders-ingenieurs.fr (M.Z.); nassim.sebaibi@builders-ingenieurs.fr (N.S.)

[2] École Supérieure d'Ingénieurs des Travaux de la Construction (ESITC-Paris), 94110 Arcueil, France

[3] Institut de Recherche de l'ESTP, École Spéciale des Travaux Publics, 28 Avenue du Président Wilson, 94234 Cachan, France

[*] Correspondence: ali.hussan@builders-ingenieurs.fr; Tel.: +33-622525442

**Abstract:** The valorization of waste fishing trawl (WFT) and waste fishing rope (WFR) fibers in cementitious materials (CMs) has gained attention in recent years; however, the lack of information on the cutting and cleaning techniques for these fibers hinders their widespread commercial utilization. Existing research primarily relies on manual cutting, which proves to be impractical for large-scale production due to its time-consuming nature and lack of industrial feasibility. This research is a component of the VALNET project and introduces an innovative technique that utilizes the cutting mill to convert WFT and WFR into fibers to effectively overcome the constraints of earlier methodologies. By employing a rotor with blades, this apparatus enables efficient and precise cutting of WFT and WFR, eliminating the need for labor-intensive manual cutting. The sustainable cleaning of WFT and WFR was carried out utilizing rain and wind by placing them outside for a certain period of time. The advancements presented in this study provide a pathway for an efficient and scalable valorization of WFT and WFR fibers in CM. The study focused on analyzing the impact of varying fiber sizes and percentages on the mechanical properties of CM. Different sizes obtained from the cutting machine and different fiber percentages were examined to gain a better understanding of their influence. The fibers obtained by the utilization of a 20 mm sieve yield optimal outcomes, while the incorporation of fibers at a volume fraction of 0.5% yields the most favorable results. Furthermore, the study presents evidence of a noticeable rise in porosity resulting from the incorporation of WFT and WFR fibers, regardless of their size and proportion. Porosity slightly increases as the fiber length increases, but the rise in fiber proportion leads to a significantly greater increase in porosity.

**Keywords:** fishing trawl fibers; fishing rope fibers; cementitious material; sustainable solution; recycled material; fiber cutting; cleaning

## 1. Introduction

The worldwide movement towards sustainability has sparked a strong enthusiasm for transforming waste materials into useful resources, representing a significant shift in thinking that spans across all industries. Within the field of construction, cementitious materials have historically played a crucial role in the development of infrastructure. However, there is a pressing need to explore and use more sustainable alternatives. In the context described, the recognition and promotion of WFT and WFR fibers to reinforce cementitious materials presents itself as a potentially advantageous pathway, combining environmental conscientiousness with inventive approaches to production. The marine ecosystem is negatively impacted by marine detritus. A large quantity of fishing gear is discarded or lost in the ocean. This gear can be fatal for marine species [1]. More than 380,000 marine creatures per year are slain by these ghost fishing gears [2]. This rising amount of marine debris is one of the most pressing issues that must be addressed to

reduce waste and preserve natural resources for a more efficient circular economy. Plastic, such as polypropylene, polyethylene, and polyethylene terephthalate, accounts for the majority of marine environment waste [3,4]. For the reuse of and reduction in marine waste, the development of efficient and environmentally friendly recycling technologies for plastic waste such as abandoned fishing trawls and ropes is crucial. Reusing discarded fishing trawl as a fiber reinforcement in cementitious composites could be one such efficient practice [5–7]. WFT and WFR have various applications, including carpet tiles, bicycle seats, chair and baggage castors, tool handles, and electrical components [8,9], as well as non-food contact bottles [10], while its employment in civil engineering is rather recent. Several studies [11–14] have described the utilization of discarded WFT fibers for the reinforcing of concrete. Concrete is well-known for its brittle nature. As stated in multiple studies [15–18], different types of randomly distributed fibers can be used to reinforce concrete by preventing or controlling the initiation and propagation of micro- or macro-cracks. The pursuit of sustainable construction is characterized by the conversion of waste materials into practical building components. The widespread utilization of WFT and WFR in the fishing industry frequently leads to their disposal at the end of their functional lifespan, hence worsening the environmental burden [19,20]. Nevertheless, new investigations have brought to light a distinctive opportunity: the incorporation of this discarded fishing gear in different applications, particularly for reinforcement purposes in cement-based materials [13,21]. This undertaking, despite its considerable promise, encounters a crucial obstacle—the proficient and productive processing, cleaning, and cutting of these fibers. The conventional approaches, which heavily rely on hand cleaning and cutting, suffer from inefficiency and impracticality when implemented on a larger scale for industrial manufacturing.

Spadea et al. [22] and Park et al. [23] demonstrated that WFT fibers (cut by hand to a length between 30 and 60 mm and diameter between 1 and 1.5 mm) could enhance the post-cracking performance of cementitious composites. In 2017, Orasutthikul and colleagues [5] conducted a comparative analysis of nylon fiber obtained from WFT and polyvinyl alcohol (PVA) fibers. The fibers were manually cut to achieve a diameter of 0.35 mm and a length of 40 mm. The inclusion of these two kinds of fiber resulted in a reduction in the compressive strength ($R_c$) of the mortar, but concurrently led to enhancements in its flexural strength ($R_f$) and tensile strength ($R_{it}$). Additionally, the fibers exhibited improvements in the post-cracking behavior of the concrete. Park et al. 2020 [7] manually cut the WFT to a diameter between 0.45 and 1.5 mm and a length of 40 mm to use them in mortar. They observed a decrease in $R_c$ as the fiber percentage increased. The addition of 1% fiber decreased the $R_c$ by up to 10% while increasing the $R_{it}$ by 32% in comparison to the controlled mortar. There was an increase in ductility under tensile load and bending loads with the increase in fiber percentage. Most of the previous studies have manually cut the WFT and WFR into fiber (Table 1).

**Table 1.** Summary of the literature review (fibers obtained with a manual cutting method).

| Authors | Control Mix $R_c$ (MPa), $R_c$ (MPa), Aspect Ratio, Fiber Dosage (%). | Control Mix $R_{it}$ (MPa), Max $R_{it}$ (MPa), Aspect Ratio, Fiber Dosage (%). | Control Mix $R_f$ (MPa), Max. $R_f$ (MPa), Aspect Ratio, Fiber Dosage (%). |
|---|---|---|---|
| [24] | 66.05, 68.55, 45, 1.0 | - | 5.63, 7.97, 77, 2.0 |
| [25] | 43.18, 41.60, 173, 1.0 | 3.77, 4.44, 173, 3.0 | - |
| [26] | 40.12, 40.74, 26.7, 1.0 | - | 3.06, 2.74, 26.7, 1.0 |
| [12] | 32.70, 24.00, 83.3, 1.0 | - | 4.77, 4.37, 83.3, 1.0 |
| [5] | 65.70, 52.50, 57, 1.0 | - | 4.80, 6.80, 57, 1.0 |
| [22] | 51.60, 44.80, 77, 1.5 | - | 4.46, 5.87, 77.0, 1.0 |
| [23] | 71.90, 61.10, 40, 1.0 | 1.53, 1.68, 40, 1.0 | - |

The workability of CM is an important parameter and should be discussed, as fiber incorporation has a direct impact on the workability of CM. With increased cohesive

forces, the slump flow or workability of concrete may decrease, depending on the type and quantity of fibers [27,28].

Spadea et al. [22] added 1.5% WFT fibers and observed a 15% decrease in $R_c$ of concrete compared to control concrete. Few other studies [11,25,26,29,30] reported the same trend in their research. On the other hand, Park et al. [23] observed a slight increase in $R_c$ with the addition of fibers. Some authors [31] have reported that the addition of fibers has no effect on the $R_c$. Therefore, the effects of fibers on the $R_c$ of concrete need to be thoroughly clarified. On the other hand, almost all previous studies have reported an increase in $R_f$ [5,12,22,24,26] and $R_{it}$ [23,25] with the inclusion of fibers. But the proportion of fiber plays a crucial role here; up to a certain proportion of fiber, $R_f$ and $R_{it}$ increase, and with a further increase in fiber dosage, there is a decrease in strength due to its balling problem and uneven distribution [5]. It is recommended to keep the fiber percentage below 2%. Most studies over the past 5 years have used a fiber percentage of less than 2% and reported an increase in $R_f$ and $R_{it}$ of fiber-reinforced CM.

WFT and WFR are typically made of high-density polyethylene or polypropylene, which are exceptionally durable and non-biodegradable substances. According to PEW's [32] "Breaking the plastic wave", the annual quantity of marine plastic litter will increase from 11 million metric tons in 2016 to 29 million metric tons in 2040. The circular economy needs to discover viable recycling solutions to reduce the environmental impact of these wastes.

*Research Significance*

Recycling and waste recovery are the foundations of the circular economy. This study is part of a project named VALNET that deals with the recycling of WFT and WFR to reinforce cementitious composites, stemming from a variety of concerns.

- As a means of mitigating environmental impacts, there is a vehement demand for recycling solutions for WFT and WFR.
- The mechanical behavior of recycled fiber-reinforced cementitious material, particularly recycled fiber from WFT and WFR, is still not very well-known.
- Most studies have focused on the mechanical properties of fiber-reinforced CM. Few discuss the cutting of these trawls and ropes into fibers.
- The increasing number of deteriorated concrete structures generates significant interest in the use of recycled fiber for concrete structure repair.

The objectives of this research are as follows:

- Define the cleaning and cutting protocol for transforming WFT and WFR into fibers with an industrial perspective.
- To analyze the physical and mechanical properties of the WFT and WFR fibers before incorporating them in the CM.
- To study the use of polyethylene fibers obtained from WFT and polypropylene fibers from WFR for reinforcing the CM.
- Comprehend the impact of both kinds of fiber on the mechanical properties of CM.
- Determine the impact of fiber size and proportion on the characteristics of mortar in both its fresh and hardened states.

## 2. Materials and Methods

### 2.1. Materials

The WFT and WFR utilized in this project are sourced from "Synergie Mer Et Littoral" (SMEL), a proficient fishing industry organization located in the Normandie region of France. Figure 1 presents the visual representations of WFT and WFR.

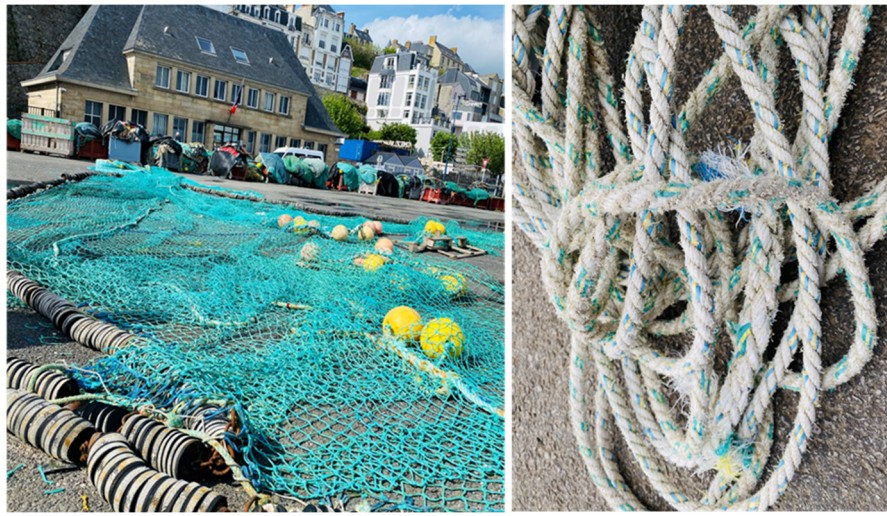

**Figure 1.** The visuals of waste fishing trawl (**left**) and waste fishing rope (**right**).

Table 2 illustrates the physical as well as mechanical properties of WFT (polyethylene) and WFR (polypropylene) fibers. Density or volume mass was calculated following NF EN ISO 18753, 2017, while $R_{it}$ and Young's modulus was calculated following international standard ASTM C1557-03, 2013.

**Table 2.** Physical and mechanical properties of WFT and WFR fibers.

| Properties | WFT | WFR |
|:---:|:---:|:---:|
| Volume Mass (g/cm$^3$) | 1.09 | 0.94 |
| Tensile strength (MPa) | 203 | 148 |
| Young's modulus (GPa) | 3.20 | 1.89 |
| Average length (mm) | 3.67, 8.67, 9.81, 15.81, 19.70 | 3.81, 7.22, 10.44, 14.98, 20.57 |
| Diameter (mm) | 100–300 | 60–200 |

The mortar was prepared using the CEM V/A 32.5 N cement, which is a type of low-clinker cement, in combination with quarry sand. The composition of the oxides of cement and mortar formulation are provided in Tables 3 and 4, respectively.

**Table 3.** Chemical composition of cement (CEM V/A 32.5 N).

| SiO$_2$ | Al$_2$O$_3$ | CaO | MgO | SO$_3$ | K$_2$O | Na$_2$O | P$_2$O$_6$ | Other |
|:---:|:---:|:---:|:---:|:---:|:---:|:---:|:---:|:---:|
| 30% | 9.8% | 46% | 2.6% | 3.0% | 1.03% | 0.32% | 0.31% | 6.94% |

**Table 4.** Mortar formulations utilizing different fiber proportions and fiber lengths.

| Cement | Sand | Water | Superplasticizer | Fibers | | Mortar Reference | |
|:---:|:---:|:---:|:---:|:---:|:---:|:---:|:---:|
| | | | | Proportion (%) | Average Length (mm) | | |
| 1 | 1.5 | 0.40 | 0.001 | 0 | 0 | MC | |
| | | | | | WFT | WFR | WFT | WFR |
| | | | | 0.3, 0.5 or 1 | 3.67 | 3.81 | MWFT4 | MWFR4 |
| | | | | 0.3, 0.5 or 1 | 8.67 | 7.22 | MWFT8 | MWFR8 |
| | | | | 0.3, 0.5 or 1 | 9.81 | 10.44 | MWFT10 | MWFR10 |
| | | | | 0.3, 0.5 or 1 | 15.81 | 14.98 | MWFT20 | MWFR20 |
| | | | | 0.3, 0.5 or 1 | 19.70 | 20.57 | MWFT-WO | MWFR-WO |

The sand used in the study is Granular Class 0/4, as per Norm NF P18-545, 2021. The sieve analysis curve for this sand is given in Figure 2.

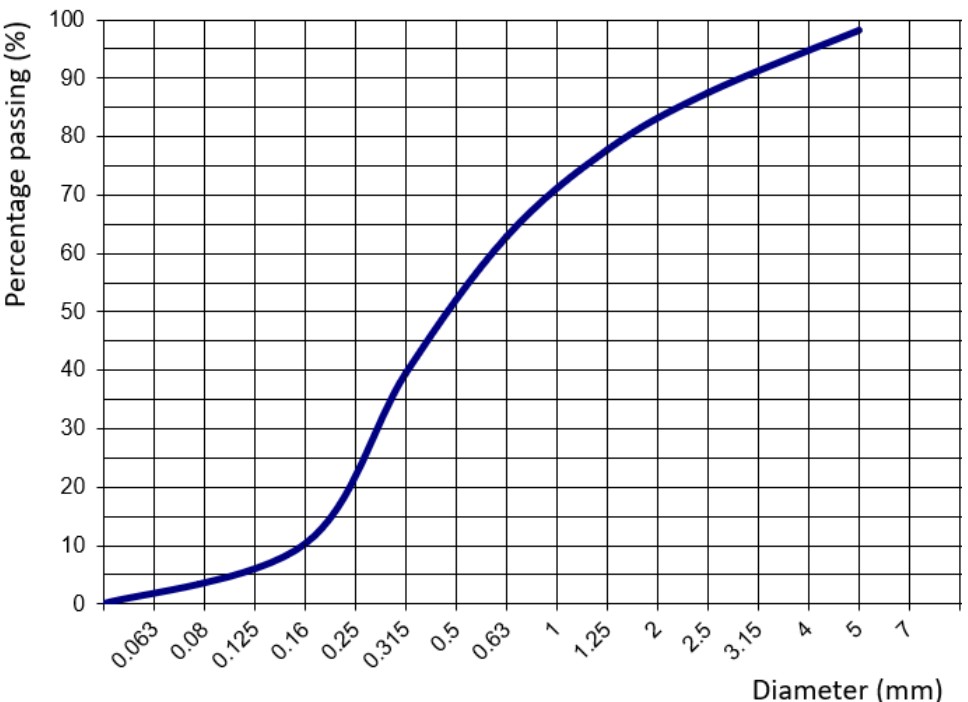

**Figure 2.** Sieve analysis curve for the quarry sand used in the study.

*2.2. Methods*

The fishing gear delivered by SMEL has a service life exceeding 10 years. By spending a significant amount of time in saltwater, the surface of WFT and WFR is likely to accumulate a substantial amount of salt. This salt accumulation may negatively impact the mechanical performance of CM [33]. Manual cleaning is a laborious and sluggish process, while machine cleaning requires a significant amount of water and power. As an example, a typical 8 kg washing machine utilizes around 60 to 70 L of water and 0.1 to 0.3 kWh of power over a period of 3 h. In order to circumvent the laborious and resource-intensive approach, we have used a sustainable technique that harnesses natural elements such as rain and wind to cleanse these fishing tools. The WFT and WFR were exposed to rain and wind for a certain duration, and subsequent salinity tests were conducted to determine the salt content in them.

The second step is to cut these fishing gears into fibers, keeping in mind the industrial point of view. This article has tried to define the protocol for efficiently cutting WFT and WFR into fibers at a large scale using a cutting machine (see Figure 3). This machine, featuring RES technology and a powerful 3 kW drive with high torque, is a high-performance cutting apparatus ideal for cutting soft, medium-hard, tough, elastic, fibrous, and heterogeneous materials. It offers variable speed control from 100 to 3000 RPM and a range of bottom sieves with aperture sizes from 0.25 to 20 mm, ensuring precise fiber size.

The workability of controlled mortar (MC) and mortar incorporating WFT (MWFT) and WFR (MWFR) fibers was measured using a flow table following French standard NF EN 1015-3, (1999). Consistency is a parameter that quantifies the viscosity and moisture content of newly mixed mortar. It indicates the ability of the fresh mortar to deform under specific stress conditions. The flow value is determined by measuring the mean diameter of a test sample of fresh mortar deposited on a defined flow table disc using a defined mold and subjected to several vertical impacts by elevating the flow table and allowing it to fall freely through a given height.

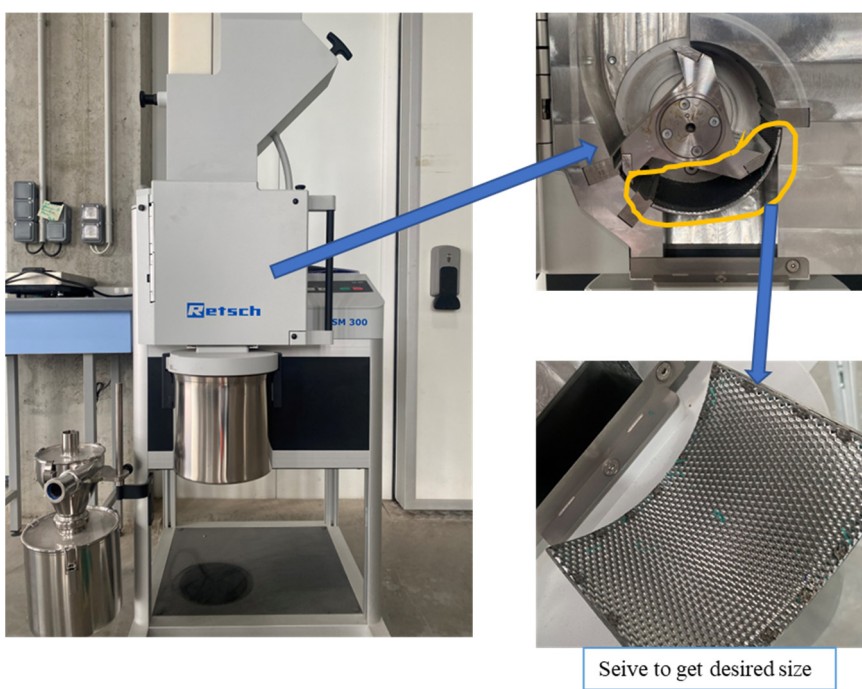

**Figure 3.** Cutting machine, utilized to mechanically cut WFT and WFR into fibers of different sizes.

The compressive strength ($R_C$) tests were conducted on 4 cm × 4 cm cubical samples using a compression test apparatus (NF EN 196-1, 2016) that was equipped with a compression force sensor capable of applying a normal load up to 250 kN with an accuracy of ±0.1% and at the rate of 2400 N/s. The determination of the uniaxial $R_C$ of a specimen can be achieved by the maximal force value (F) per area of the specimen ($A_c$) according to Equation (1).

$$R_c = \frac{F}{A_c} \tag{1}$$

The flexural strength ($R_f$) of mortar was determined by a three-point bending test (NF EN 196-1, 2016) with a machine capable of applying a load of up to 10 kN with a rate of loading 50 N/s. The $R_f$, in N/mm$^2$, was calculated using the following formula:

$$R_f = \frac{3F \times l}{2b \times d^2} \tag{2}$$

where F is the maximum load applied to the specimen, in newtons (N); l is the distance between the support rollers, in millimeters (mm); b is the width of the specimen, in millimeters (mm); d is the depth of the specimen, in millimeters (mm).

## 3. Results and Discussions

### 3.1. Optimization of Cleaning and Cutting of Fibers

WFT and WFR were placed outside under the effects of rain, wind, and temperature for a period of six months (Figure 4).

The rainfall that flowed through the WFT and WFR was collected at the bottom, and salinity measurements in parts per thousand (ppt) were conducted on the collected water every fifteen days. As shown in Figure 5, the salinity of the accumulated water decreased over time. The meteorological data for the city of Caen in the French province of Normandie may be obtained from the following webpage: https://weatherandclimate.com/ (accessed on 30 April 2024).

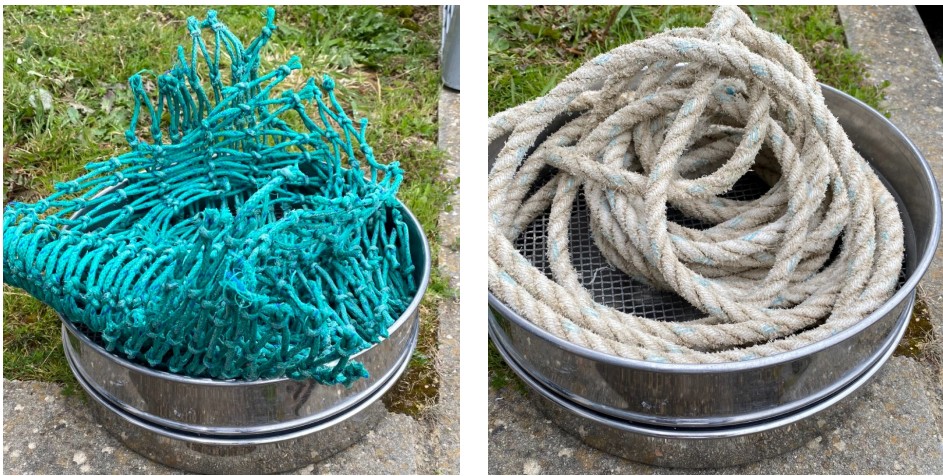

**Figure 4.** Experimental setup to collect rainwater that passed through the WFT and WFR.

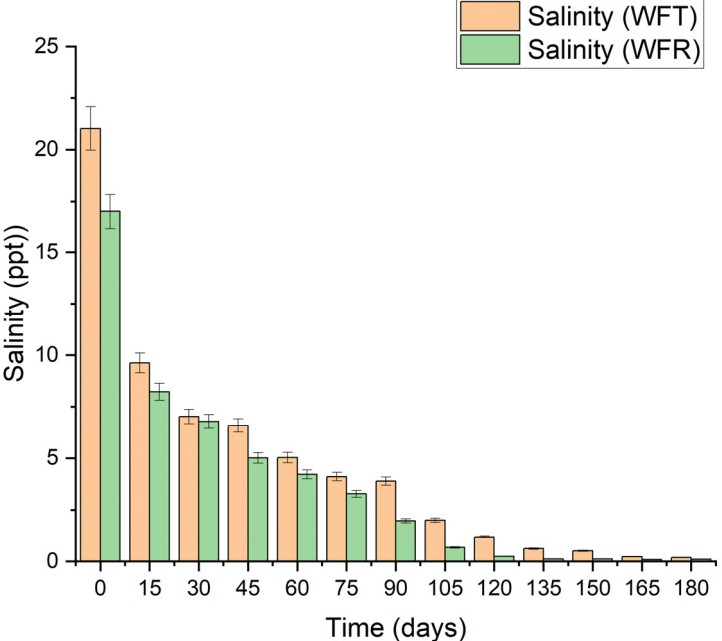

**Figure 5.** The salinity measurement of WFT and WFR under the effects of rain and wind over the period of six months from August 2023 to the end of January 2024.

The cutting machine has the capability to divide the WFT and WFR into various dimensions by using different sieves positioned at the end of the cutting blade. The apparatus came with four sieves measuring 4 mm, 8 mm, 10 mm, and 20 mm, respectively. The machine also possesses the capacity to cut the fiber without employing a sieve; however, this method yields fibers of varying lengths ranging from 5 mm to 45 mm. The final output for WFT and WFR fibers is depicted in Figure 6.

The analysis of fiber size distribution was conducted using a software application called ImageJ. Two hundred randomly selected fibers were placed on a white piece of paper for each sieve size to capture clear images. The image was further analyzed using ImageJ software (Java 1.8.0_271 (64-bit)) to obtain the size distribution of the fibers, as shown in Figure 7 for WFT fibers and Figure 8 for WFR fibers.

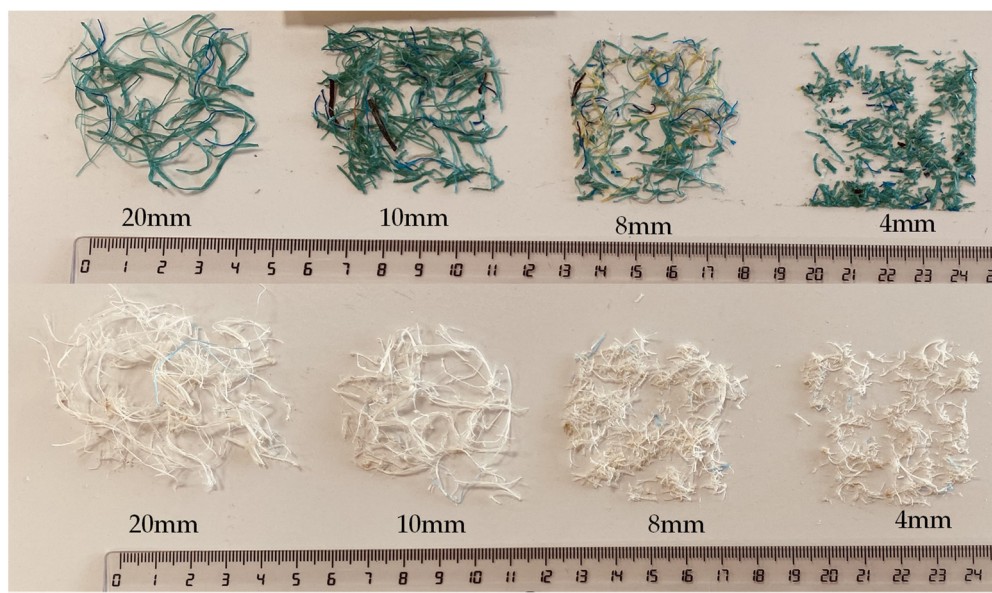

**Figure 6.** Different sizes of fibers were obtained using a mechanical cutting machine from WFT (**top**) and WFR (**bottom**).

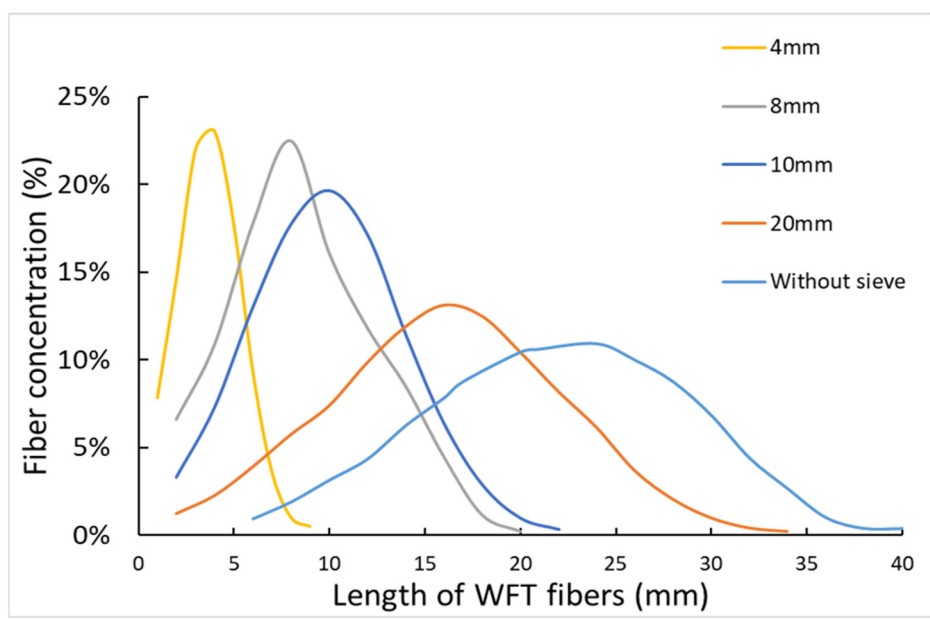

**Figure 7.** Size distribution of WFT fibers obtained for each sieve size.

There is a significant variation in the size of fibers recovered from each sieve. However, the size of each fiber is primarily centered around its corresponding sieve size. For instance, when employing a 20 mm sieve, the concentration of fibers primarily occurs within the vicinity of 20 mm, indicating that the majority of fibers had a length about equal to 20 mm. In a similar manner, the use of 10 mm, 8 mm, and 4 mm sieves results in the concentration of fiber length around their corresponding sieve sizes. It is important to highlight that a lower sieve size corresponds to a narrower curve, indicating a reduced fluctuation in fiber length.

The measurement of electrical energy consumption is conducted for each output size, as depicted in Figure 9. It should be noted that the aforementioned energy consumption pertains to the process of cutting 1 kg of WFT or WFR.

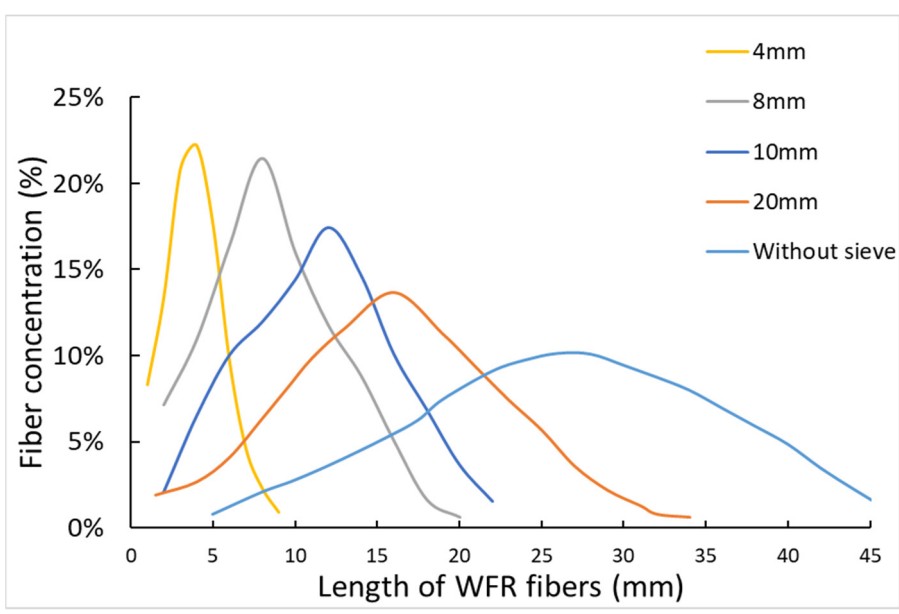

**Figure 8.** Size distribution for WFR fibers obtained for each sieve size.

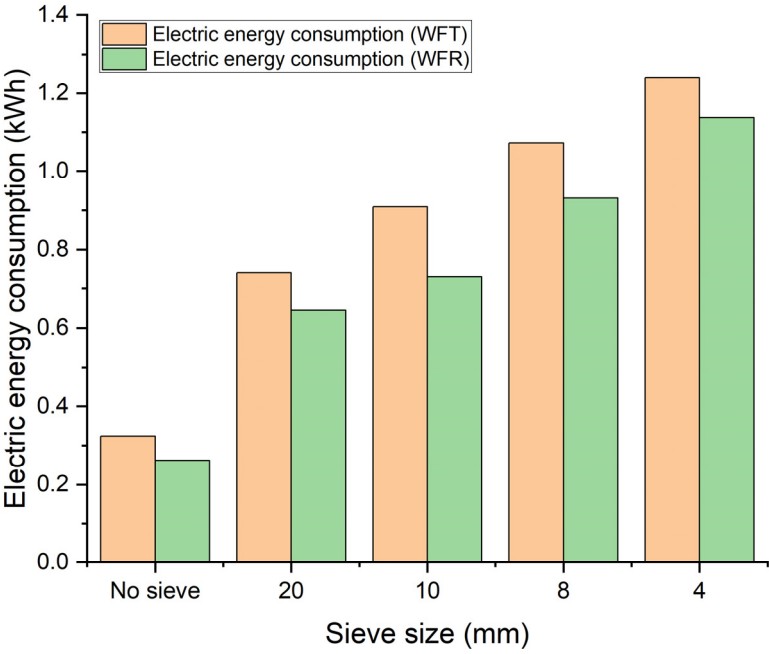

**Figure 9.** Electrical energy consumption to obtain fibers utilizing sieves of different sizes.

In relation to production efficiency, no instances of material waste are observed during the process of cutting the WFT and WFR using this apparatus to extract fibers. The energy consumption required to obtain 1 kg of material varies depending on the size, resulting in varying time duration to produce 1 kg of fibers. Table 5 presents the duration, measured in minutes, necessary for the cutting of 1 kg of fibers, together with their corresponding average size.

The use of the mechanical cutting machine for fiber cutting demonstrates encouraging outcomes. The subsequent phase involves comprehending the application of WFT and WFR fibers within the CM.

**Table 5.** Electricity consumption, the time required to cut 1 kg of fibers, and their average length.

| Sieve Size | Electricity Consumption (kWh) | | Time Required (Minutes) | | Average Length (mm) | |
|---|---|---|---|---|---|---|
| | WFT | WFR | WFT | WFR | WFT | WFR |
| 4 mm | 1.24 | 1.21 | 25.85 | 23.58 | 3.67 | 3.81 |
| 8 mm | 1.07 | 0.98 | 22.48 | 19.45 | 8.67 | 7.22 |
| 10 mm | 0.91 | 0.62 | 19.20 | 15.42 | 9.81 | 10.44 |
| 20 mm | 0.74 | 0.44 | 15.87 | 13.73 | 15.81 | 14.98 |
| Without sieve | 0.32 | 0.26 | 7.52 | 6.08 | 19.70 | 20.57 |

### 3.2. Workability of Mortar Incorporating Fibers

The investigation into the impact of fiber length (five different lengths) and volume percentage (0.3%, 0.5%, and 1%) on the performance of CM revealed intriguing insights. Mortar formulations, designated as MC (without fibers), MWFT4, MWFT8, MWFT10, MWFT20, and MWFT-WO (made with WFT fibers obtained using 4 mm, 8 mm, 10 mm, and 20 mm sieves and without sieve, respectively), as well as MWFR4, MWFR8, MWFR10, MWFR20, and MWFR-WO (containing WFR fibers obtained using 4 mm, 8 mm, 10 mm, and 20 mm sieves and without sieve, respectively), were subjected to thorough analysis.

The findings of the flow table for mortar, including fiber proportions of 0.3%, 0.5%, and 1% by volume, are presented in Figures 10–12, respectively.

Previous studies [28,29,34,35] have already revealed a decrease in the workability of mortar with fiber inclusion. The workability is affected by the length, surface morphology, and surface area of fibers [36]. Our study confirms and supports this observed trend. The results indicate that the length of the fibers has a minimal impact on the workability of mortar; however, the proportion of fibers has a more significant effect. When the volume of fibers is constant, it implies that the mass also remains constant. This correlation can account for the minimal impact of fiber length on the workability of the mortar. However, a marginal decrease in workability was noted when the length of fibers increased. A rise in percentage results in a drop in workability, and the addition of 1% of fibers demonstrates the lowest level of workability. When comparing WFT (polyethylene) and WFR (polypropylene), the inclusion of WFR fibers results in a greater decrease in workability compared to WFT fibers. The increased water absorption coefficient of WFR fibers may be the reason for this relatively higher decrease in workability.

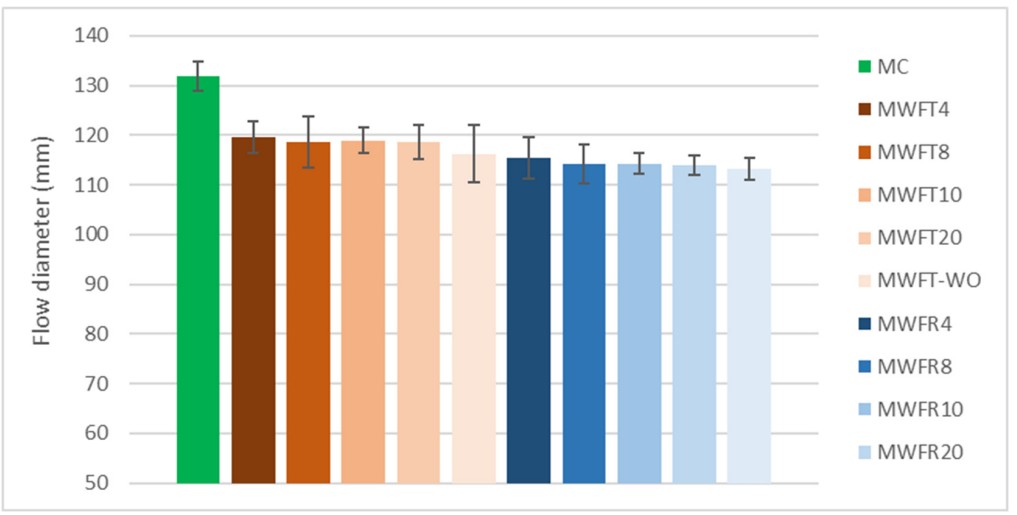

**Figure 10.** Workability of mortar, measured using a flow table for each size (the proportion for each fiber length included is 0.3% by volume).

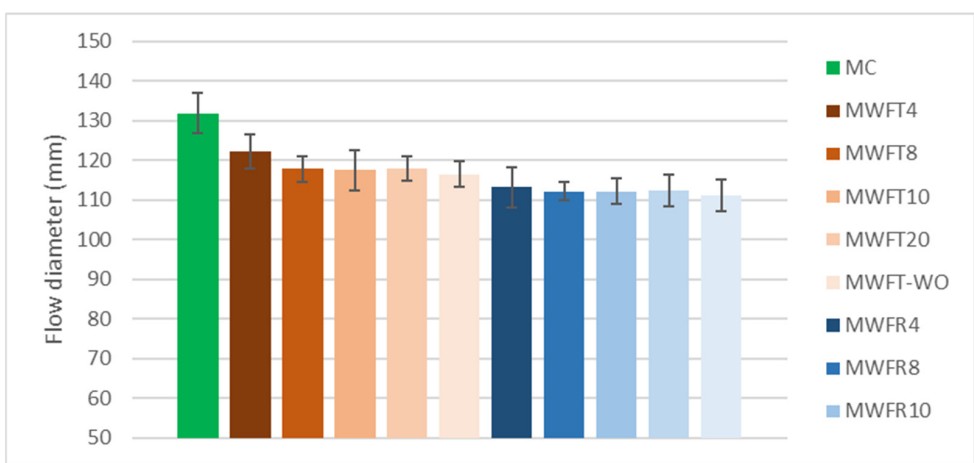

**Figure 11.** Workability of mortar, measured using a flow table for each size (the proportion for each fiber length included is 0.5% by volume).

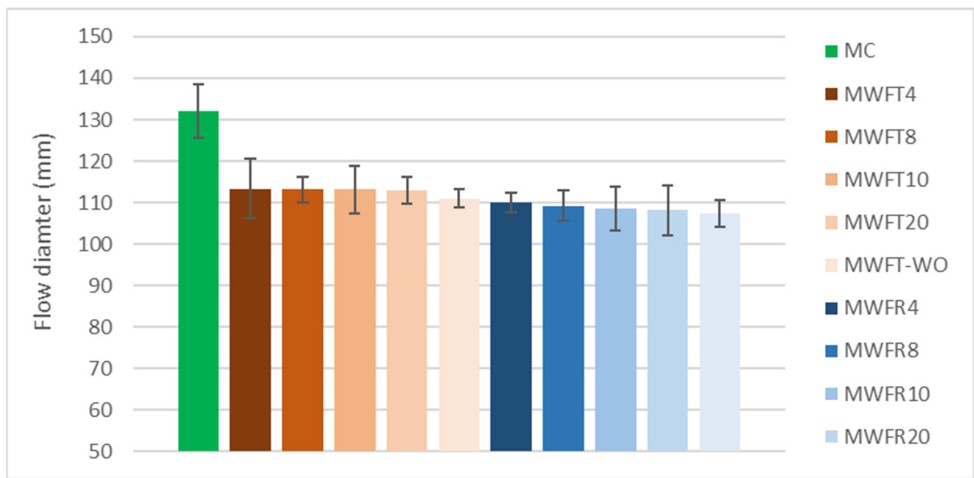

**Figure 12.** Workability of mortar, measured using a flow table for each size (the proportion for each fiber length included is 1% by volume).

### 3.3. Mechanical Properties in Hardened State

The determination of water-accessible porosity is conducted in accordance with the French standard NF P18-4592, 2022. The controlled mortar samples and the samples fabricated using WFT and WFR fibers were inserted into a container linked to vacuum equipment. The air was removed from the container until the pressure within reached 25 mbar or below. The pressure was sustained for a duration of four hours prior to being filled with water. The three types of samples were immersed in this water for a duration of 44 h. The mass of each sample was determined in water ($M_{water}$) using a hydrostatic balance, in air ($M_{air}$) using a normal balance, and after drying in an oven to obtain the dry mass ($M_{dry}$). The measurement of water-accessible porosity was conducted using Equation (3).

$$Porosity = \frac{M_{air} - M_{dry}}{M_{air} - M_{water}} \tag{3}$$

The fiber proportion was maintained at 0.5% by volume, and the impact of length increase on water-accessible porosity of mortar incorporating WFR fiber (MWFT) and WFR fiber (MWFR) is shown in Figure 13.

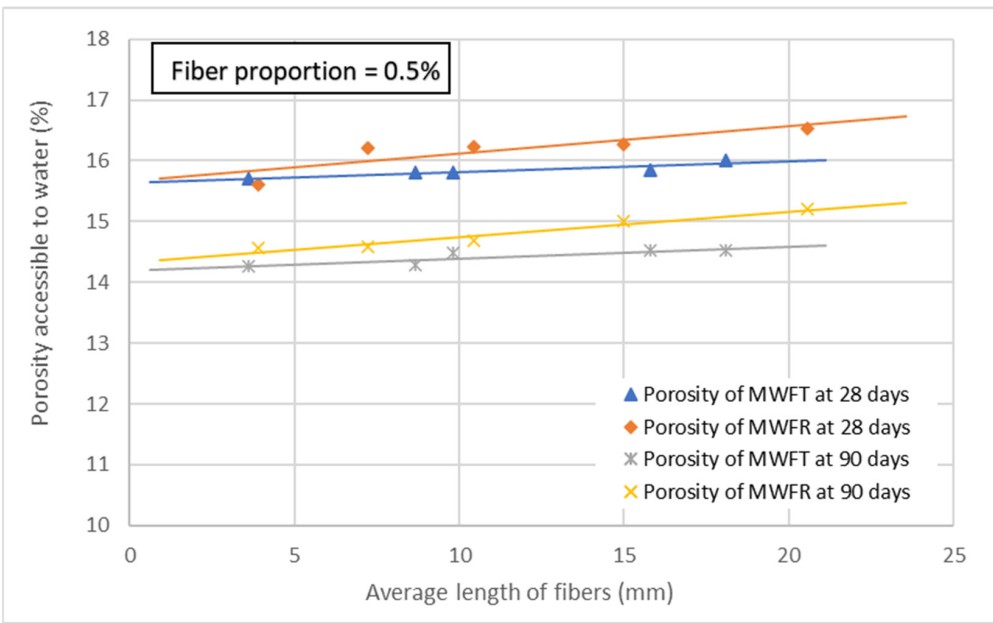

**Figure 13.** Effect of fiber length on the water-accessible porosity of mortar.

The study initially focused on how the length of fibers impacts the porosity of the mortar. The results clearly show a small increase in porosity as the fiber length increases. MWFR exhibits greater porosity values at the specified lengths at both 28-day and 90-day curing periods compared to MWFT. Porosity has been observed to decrease at 90 days when compared to porosity at 28 days.

The subsequent objective is to determine how the volume of fibers influences the porosity of mortar while keeping the length of fibers constant (WFT-20). The porosity of mortar is significantly affected by the fiber proportion, as a noticeable rise in porosity is observed with an increase in fiber proportion. The porosity of mortar is 14% at 0% fiber volume and increases to 17.23% at 1% fiber volume for MWFT-20 and 17.80% for MWFR-20 after a 28-day curing period (Figure 14). Several studies have already documented an increase in the porosity of cementitious materials with the inclusion of fibers [36–39].

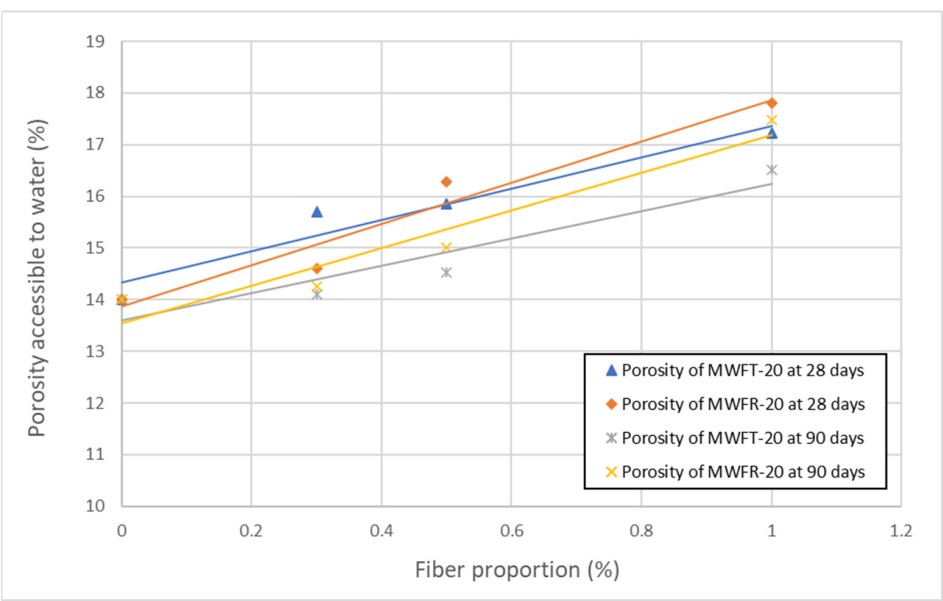

**Figure 14.** Effect of fiber proportion on the water-accessible porosity of mortar.

Now, to measure three-point $R_f$ and $R_C$, prismatic specimens with dimensions of 4 cm × 4 cm × 16 cm were prepared as per NF EN 196-1, 2016. A total of eighteen samples were prepared, consisting of six samples using WFT fibers, six samples incorporating WFR fibers, and six samples without any fiber inclusion for each proportion and each length of the fiber. These samples were tested following 7-day and 28-day curing periods in water at room temperature. Figures 15 and 16 give the evolution of RC at 7 and 28 days for MC, MWFT, and MWFR.

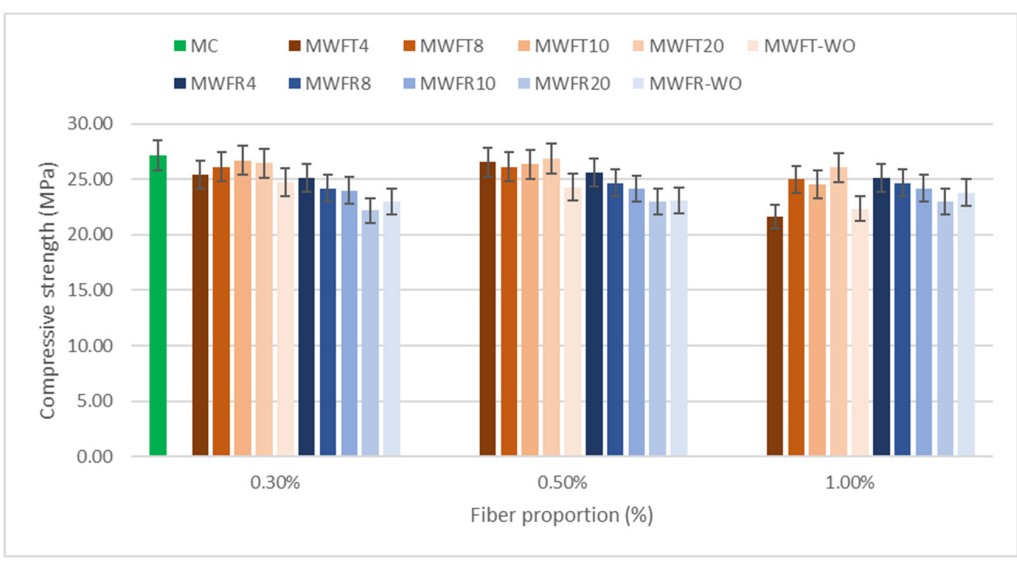

**Figure 15.** Compressive strength test results after 7 days of curing in water for controlled mortar, and mortar incorporating WFT fibers and WFR fibers.

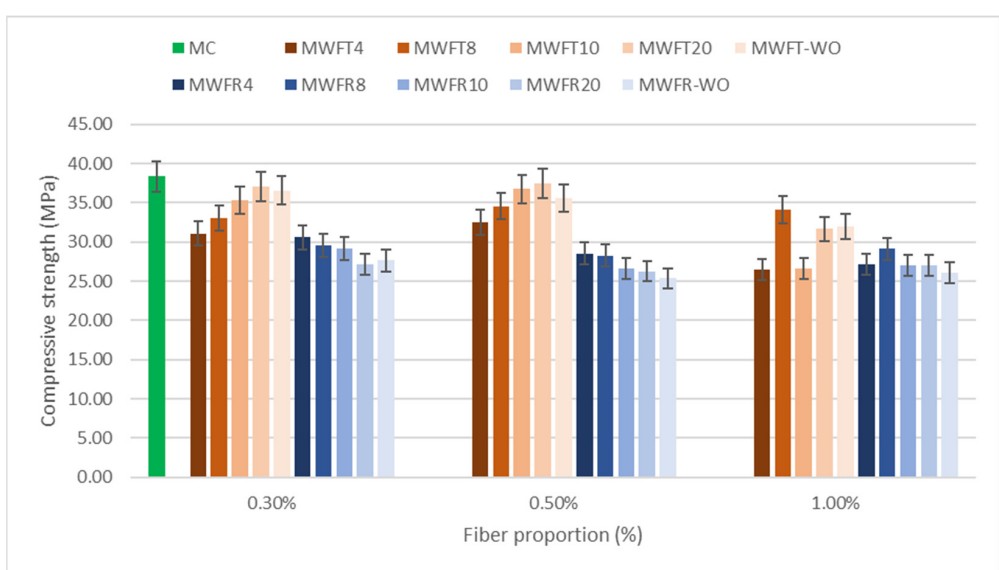

**Figure 16.** Compressive strength test results after 28 days of curing in water for controlled mortar, and mortar incorporating WFT fibers and WFR fibers.

The value of compressive strength presented in Figures 15 and 16 is the average value of three separate samples. After 7 days of curing in water, both MWFT and MWFR showed a slight decrease in compressive strength compared to MC. A significant decrease in compressive strength was seen after 28 days of curing in water when either WFT or WFR fibers were added to the mortar. WFR fibers exhibited a greater decrease in compressive strength than WFT fibers. When discussing various fiber lengths, it is noted that fiber

length does not significantly impact the compressive strength of CM. However, a slight drop is observed as the fiber length increases. Similarly, a rise in the proportion of fibers led to a modest drop in the compressive strength of CM. After 7 days of curing in water, MWFT4 experienced the greatest decrease in compressive strength, at 20.44%, with a 1% fiber volume, while MWFT20 had the smallest reduction, at 1.01%, with a 0.5% fiber volume. After 28 days of curing in water, the MWFR-WO sample with 1% fiber volume exhibited the highest drop in compressive strength, at 31.90%, whereas the MWFT-20 sample with 0.5% fiber volume showed the lowest reduction, at 2.26%. Figures 17 and 18 give the evolution of flexural strength at 7 and 28 days for MC, MWFT, and MWFR.

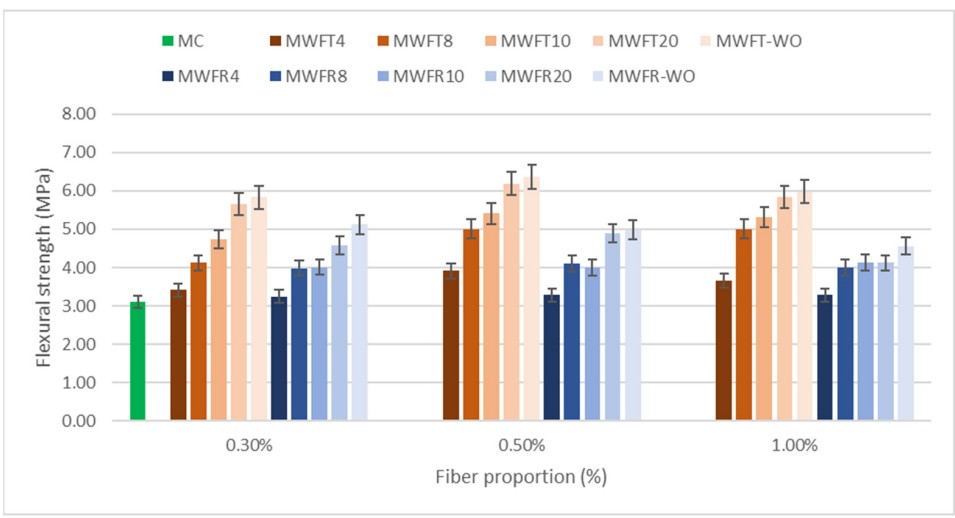

**Figure 17.** Flexural strength test results after 7 days of curing in water for controlled mortar, and mortar with WFT fibers and WFR fibers.

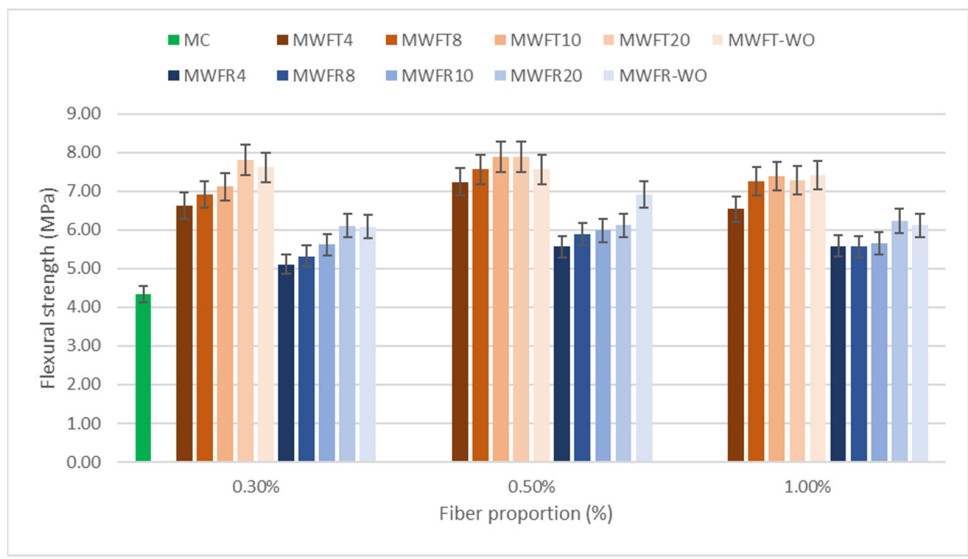

**Figure 18.** Flexural strength test results after 28 days of curing in water for controlled mortar, and mortar with WFT fibers and WFR fibers.

The study showed a notable increase in flexural strength at 7 days and 28 days of curing, as seen in Figures 17 and 18. Adding WFT and WFR fibers at a volume fraction of 0.5% significantly boosted flexural strength when compared to 0.3% and 1% fiber volumes. A loss in strength above 0.5% fiber volume could be due to a balling problem and uneven fiber distribution [5].

After 28 days of curing in water, the MWFR20 samples containing fibers collected via a 20 mm sieve (with an average length of 15.81 mm) exhibited the most favorable result, with an 81.79% enhancement in flexural strength. Therefore, it can be inferred that fibers produced by cutting WFT (polyethylene) using a 20 mm sieve yield optimal outcomes when incorporated into the mortar at a volume of 0.5%. Fiber derived by cutting WFR (polypropylene) demonstrates favorable outcomes as well. However, their performances are slightly inferior when compared to WFT fibers. The significant rise in flexural strength is apparent based on previous studies. Orasutthikul et al. [5] observed a significant 41.66% enhancement in the mortar when a 20 mm fiber with a volume fraction of 1.5% was included. Furthermore, Ref. [22] documented a 35% rise, Ref. [6] documented a 41% rise, and Ref. [24] showed a 41.56% rise in flexural strength as a result of incorporating fibers.

Past research has provided little information about the valorization of waste fishing rope (WFR). Waste fishing trawl (WFT) valorization has been the subject of the majority of prior research. Hence, comparing the results of WFT and WFR is challenging, and we suggest conducting more research to have a comprehensive understanding of the contrasting mechanical performances of WFT fibers and WFR fibers. The superior mechanical performance of WFT over WFR in our situation may be attributed to the higher density, greater tensile strength, and lower water absorption of WFT in comparison to WFR.

The results are consistent with Karahan and Atis's [23] findings, indicating that adding fibers to the cement matrix can create more voids, leading to a decrease in the compressive strength of the material. This decrease in compressive strength aligns with observed patterns in other research investigations. The increase in flexural strength observed is also consistent with prior findings mentioned in the introduction. The results highlight the significance of taking into account fiber length and volume proportion in CM formulations and offer significant insights for enhancing mechanical performance.

The MWFT-20 and MWFR-20 with 0.5% fiber volume have yielded the most optimal mechanical outcomes. In order to enhance the comprehension of these two categories of fiber-reinforced cementitious material, mid-span deflection curves and toughness indices were formulated. The experiment involved conducting three-point flexural tests on three samples, with each having dimensions of 4 cm × 4 cm × 16 cm. It was visually observed that MC broke into two pieces after the application of maximum force, while MWFT-20 and MWFR-20 sustained the load and did not split entirely into two (Figure 19).

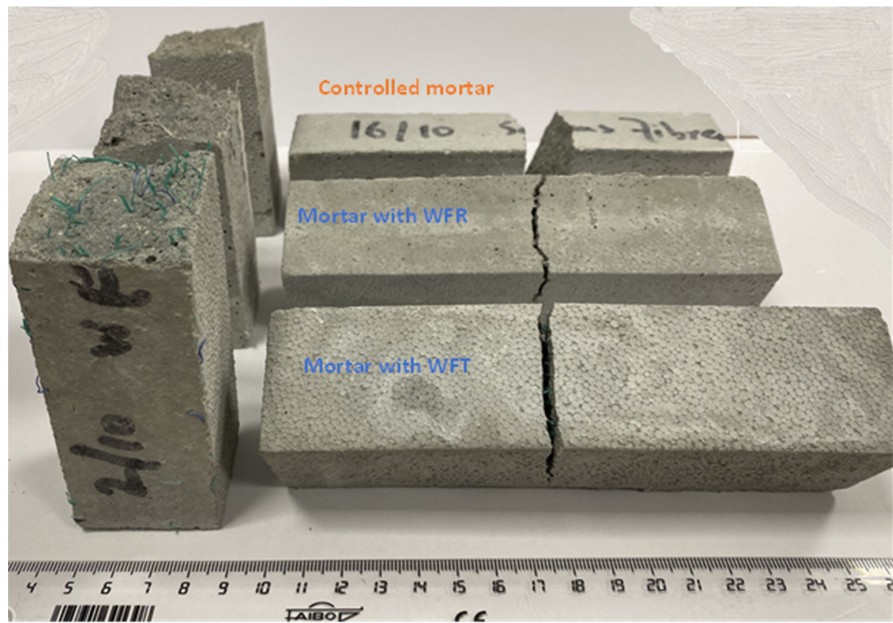

**Figure 19.** Samples after the application of maximum flexural load.

Mid-span deflection curves and toughness indices are developed specifically for the MWFT-20 and MWFR-20 (0.5% fiber volume) to be compared with controlled mortar (MC). Since these formulations have yielded the best results, mid-span deflection curves and toughness indices are produced to gain a deeper understanding of their flexural behavior. Figure 20 illustrates the load–mid-span deflection curves for each of the three formulations. When making a comparison between MC, MWFT-20, and MWFR-20, it becomes apparent that the latter two demonstrate a higher peak load. The MC exhibits a brittle failure mode characterized by a sudden fall in applied load to zero following the peak load. On the other hand, it can be observed that when using WFT and WFR fibers, the mortar has the ability to transfer stress even after the formation of cracks and reaching maximum load. This is evident from the steady decrease in load experienced after reaching the maximum load and has been reported by previous research [26,40,41].

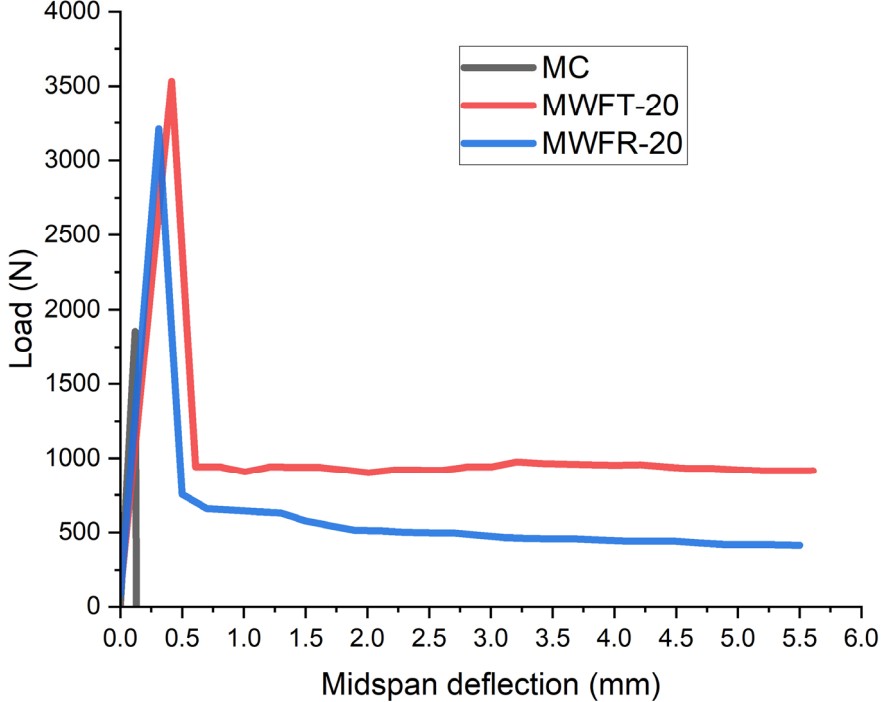

**Figure 20.** Load–mid-span deflection curve for controlled mortar and mortar prepared using WFT-20 and WFR-20 fibers.

The toughness of a test specimen can be determined by measuring the areas under the load-deflection curve. This measurement reflects the specimen's ability to absorb energy and is influenced by its geometric properties and the way it is loaded. The test method reported by ASTM 1018-97 [42] evaluates the toughness properties of fiber-reinforced CM by measuring the area under the load-deflection curve of a beam supported at two points and loaded at a third position. This test method evaluates the toughness indices that indicate the material's behavior until it reaches the point of deflection. These indices are generated by dividing the area under the load-deflection curve up to a given deflection threshold by the area up to the first crack (ASTM 1018-97). These indices provide residual strength factors that quantify the average amount of load retained after a crack has occurred, expressed as a percentage of the load at the initial crack. ASTM 1018-97 also defines the residual strength factors $R_{5,10}$ and $R_{10,20}$ by the following equations.

$$R_{5,10} = \frac{100}{5-10} \left( I_5 - I_{10} \right) \tag{4}$$

$$R_{10,20} = \frac{100}{10-20} \left( I_{10} - I_{20} \right) \tag{5}$$

The results for toughness indices and residual strength factors are given in Table 6.

**Table 6.** Toughness indices and residual strength factors for mortar prepared using WFT-20 and WFR-20 fibers.

| Sample Type | Toughness Index | | | $R_{5,10}$ | $R_{10,20}$ |
|---|---|---|---|---|---|
| | $I_5$ | $I_{10}$ | $I_{20}$ | | |
| MWFT-20 | 2.32 | 3.55 | 6.32 | 24.61 | 27.68 |
| MWFR-20 | 2.25 | 3.21 | 4.76 | 19.07 | 15.50 |

The mortar made with both types of fibers has commendable toughness. MWFT-20 exhibits superior toughness compared to MWFR-20.

### 4. Discussion

The study's conclusions are as follows.

- The cutting mill enables accurate and efficient cutting of WFT and WFR into fibers, overcoming the constraints of manual techniques. The numerous sieve sizes of 4 mm, 8 mm, 10 mm, and 20 mm offer flexibility in collecting fibers of various lengths, enhancing the range of use.
- Analysis of fiber size distribution shows that the cutting mill generates fibers that are predominantly centered around their respective sieve sizes. Smaller sieve sizes lead to narrower curves, showing less variation in fiber length. This knowledge is crucial for comprehending and managing the characteristics of the cementitious materials produced.
- The natural cleaning process of WFT and WFR had favorable outcomes, as the salt content was effectively lowered to the target level over a period of 6 months when exposed to rain and wind. Despite the lengthy duration, the cleaning of WFT and WFR does not require any mechanical energy or manpower. Simply position them outside without any other requirements. However, mechanical cleaning remains a viable option in cases when the cleaning of WFT and WFR is time-sensitive and there is no room for delay. Nevertheless, it should be noted that this method would be inefficient and detrimental to the environment.
- Studying how different fiber sizes and proportions affect the mechanical properties of CM is a critical part of the research. The study indicates that fibers acquired through a 20 mm sieve produce the best results when added to the cementitious material, and a fiber volume fraction of 0.5% yields the most advantageous effects. By augmenting the fiber concentration, clusters of fibers may form, leading to an increase in voids within the mortar and a subsequent decrease in its density. Figure 21a depicts the distribution of fiber that was added at a volume concentration of 0.5%. In contrast, the figure on the right illustrates the development of clusters after the fiber concentration was increased to 1%, which could be the cause of the reduction in the mechanical resistance of the mortar.
- The workability of MC was affected differently by the presence of WFT and WFR fibers. The MC with WFT fibers showed a lesser reduction in workability compared to the MC with WFR fibers. Hence, the significance of fiber type in controlling the workability of the mortar is underscored. Furthermore, there was a direct correlation seen between the length of fibers and the decrease in workability. This highlights the need for conducting a comprehensive analysis of the fiber dimensions in order to properly address any possible difficulties associated with workability in CM. To address the problem of decreased workability, one might examine the selection of superplasticizer and the adjustment of the water/cement ratio. Nevertheless, using this method may lead to a decrease in mechanical characteristics. Hence, it is important to thoroughly investigate the optimal equilibrium among workability, water/cement

ratio, and the use of superplasticizers. This might provide a novel study topic for future investigation.

- The experiment shows a significant rise in porosity in CM when fibers are added. The results indicate that porosity shows a minor rise when fiber length increases, but a more significant increase is observed with larger fiber fractions. This trend highlights the balance needed between fiber proportion and porosity when optimizing the mechanical characteristics of the material.

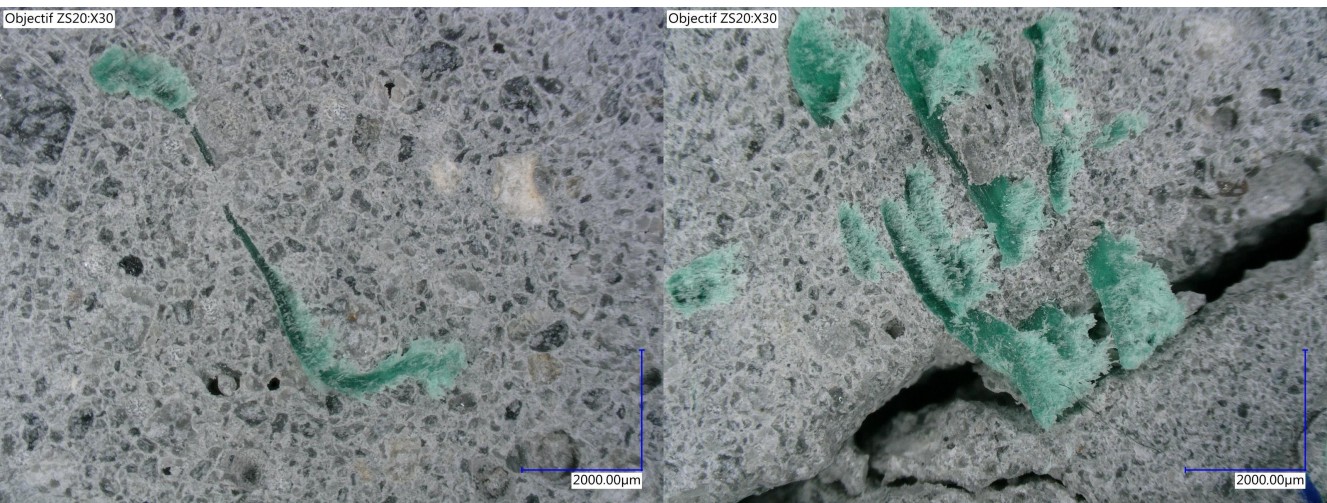

**Figure 21.** Microscopy images for fiber-reinforced CM, (**a**) CM containing 0.5% fiber concentration by volume and, (**b**) CM incorporating 1% fiber concentration.

## 5. Conclusions

The results of this research provide useful insights into the viability of using abandoned fishing equipment (WFT and WFR) as a sustainable fiber source for improving cement-based composites. The use of WFT and WFR fibers in CM has significantly improved its flexural capabilities while only slightly reducing its compressive strength, making them ideal substitutes for industrial fibers. The use of a cutting mill has been essential in overcoming the limitations of manual cutting methods, offering efficient and scalable fiber extraction. The research emphasizes the need for attaining an exact equilibrium while producing these fibers since the dimensions and ratio of fibers significantly affect the mechanical efficiency of CM. The notable increase in flexural strength, particularly seen in formulations such as MWFT-20 with a fiber volume of 0.5%, underscores the capacity of waste fibers to enhance the structural strength of cement-based composites. However, this enhancement leads to increased porosity, necessitating a meticulous approach to achieve a harmonious equilibrium between strengthening characteristics and ease of usage.

**Author Contributions:** Conceptualization, A.H. and B.E.H.; methodology and experimental work, A.H.; data analysis, A.H., N.S. and B.E.H.; writing—original draft preparation, A.H. and B.E.H.; review and editing, N.S., B.E.H. and M.Z.; supervision, N.S., B.E.H. and M.Z.; project administration, N.S., B.E.H. and M.Z. All authors have read and agreed to the published version of the manuscript.

**Funding:** The author would like to report the financial support of 547,929.74 euros over the 24 months by ADEME based on the "Contrat de Plan Interrégional Etat-Régions (CPIER VdS) la Vallée de la Seine" and the Normandy and Ile-de-France region. Please note that all the funding parties are government entities.

**Institutional Review Board Statement:** Not applicable.

**Informed Consent Statement:** Not applicable.

**Data Availability Statement:** The data presented in this study are available upon request from the corresponding author. The data are not publicly available due to the fact that all the results are reported in the internal reports of the laboratory.

**Acknowledgments:** The author would like to acknowledge the financial support of ADEME based on the "Contrat de Plan Interrégional Etat-Régions (CPIER VdS) la Vallée de la Seine" and the Normandy and Ile-de-France region. The authors would also extend their gratitude to the project partners, Aquimer, Coopération Maritime, EM Normandie, and SMEL.

**Conflicts of Interest:** The authors declare no conflicts of interest.

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
