# Peer review of "Innovative Cutting and Valorization of Waste Fishing Trawl and Waste Fishing Rope Fibers in Cementitious Materials"

_applsci, doi:10.3390/app14103985_

Round 1

Reviewer 1 Report

Comments and Suggestions for Authors

The article is related to the use of waste fishing nets in cement composites. This is an interesting way of using waste materials in construction, which should be promoted. The article hits the scope of the Applied Science journal. Below are some comments: 

1. the article needs language correction.

2. The length of the fibers that were used in the study should be better characterized. The description of “different sizes” in Tab 2 is insufficient. To ensure the scientific need for other researchers to check the results, the length information (e.g., average) should be provided. 

3. the table numbers are confused (twice tab 2).

4. the composition of the mortar used in the study should be given as kg/m3 content shown. The ratio is inaccurate.

5. Please check tab 1 (chemical composition) the sum of compounds is 93.06% - and where is the remaining 7%? The column “other” should also be added.

6. fig 2 - the horizontal axis can be limited to 6.3 mm.

7. Compressive strength and flexural strength should be described with the designation from the standard (not CS;FS). In the formulas there should be a multiplication sign, not a dot.

8.I ask that the authors discuss in the article the sense of using nets for concrete, if before using them it is necessary to reduce their salinity which takes about 100 days.

9. It is strange that the size of the error bars is the same for each of the tested series in Figures 10,11,12. 

10. Figures 10,11,12 show different test series - why is the material composition of these series not given in section 2 of this publication?

11. in Fig. 13 error: “proprtion"

12 Pattern 3 - “MAir” and “Mair” are the same thing or something different?

13.The results shown in the charts are very poorly discussed.

14. the flexural strength results shown in fig 17 are surprisingly good - please compare them with the literature in detail. PP fibers give much worse results. 

Comments on the Quality of English Language

Should be improved

Author Response

Hello, 

Please find attached a file with detailed answers to your questions as well as revised manuscript as per your guidelines. 

Sincerely,
Ali HUSSAN

Reviewer 2 Report

Comments and Suggestions for Authors

This study investigates the utilization of waste fishing trawl (WFT) and waste fishing rope (WFR) fibers in cementitious materials (CM), addressing the lack of efficient cutting and cleaning techniques for these fibers. Previous methods relying on manual cutting proved impractical for large-scale production; however, this research aims to introduce an innovative cutting technique that enables precise and efficient fiber processing, overcoming earlier constraints. By analyzing varying fiber sizes and percentages, the study demonstrates that fibers obtained through a 20 mm sieve and incorporated at a volume fraction of 0.5% yield optimal mechanical properties in CM, albeit with an increase in porosity observed regardless of fiber size or proportion.

The study, part of a project, aims to explore recycling solutions for waste fishing twine (WFT) and waste fishing ropes (WFR) to strengthen cementitious composites. Concerns arise from limited knowledge of the mechanical behavior of recycled fibers, especially from WFT and WFR, and a lack of focus on their cutting into usable fibers. Objectives include defining cleaning and cutting protocols for transforming WFT and WFR into fibers, analyzing their physical and mechanical properties, studying their use in cementitious composites, and understanding the impact of fiber characteristics on mortar properties.

The study stands out for its originality, as there's a scarcity of research on the mechanical properties of this particular type of fiber-reinforced concrete. Furthermore, its findings could significantly enhance the sustainability of construction projects by introducing viable recycling solutions for waste fishing twine and ropes.

Having said that, I would like to share my specific comments with the authors:

Figure 21 from ASTM standards might need to be reconsidered concerning potential copyright challenges.

The paper's absence of a conclusion section appears atypical and warrants attention. This critical omission detracts from the overall structure and completeness of the research, potentially hindering readers' understanding of the study's key findings and implications.

The section and subsection numbering require revision. For instance, there are currently two subsections, both labeled 3.2.

I couldn't determine the specific test or standard used to measure workability. If the slump test is employed, could you include images of the tested samples? From my analysis, it seems that incorporating the studied fibers into the concrete results in decreased workability. In light of this, what solutions can the authors propose? How can this concrete with reduced workability still be utilized in construction? Is it feasible to pump it to higher levels for concreting purposes?

It appears that the specimen used for the compressive test had dimensions of 4cm x 4cm x 16cm. This proportion seems unusual to me. Typically, the height of the specimen is twice its width to minimize the risk of geometrical instabilities.

The decreased compressive strength of fiber concrete seems puzzling to me, considering that the presence of fibers typically enhances the bond and cohesion within the concrete, thereby reducing the likelihood of various failure modes. Could you provide insights into the failure modes observed in specimens under compressive loads, and use that information to justify your experimental observations?

Did you record the strain in the concrete? Does presence of fibders increase the deformation capacity of specimens?

On Line 347, the authors attributed the reduced tensile strength in specimens with higher fiber densities to the "balling effect." Firstly, I believe "balling" is the correct spelling. Secondly, could you offer additional evidence, such as images, to further support your explanation? Moreover, does the balling effect alter the failure mode of the specimens under flexural loadings?

Could you provide insights into the impact of the studied fibers in the concrete on "element size effects"? Further details can be found at: https://doi.org/10.1016/j.engfracmech.2021.108193

References look adequate to me. A more detailed discussion of the results and a comparison with the available studies is needed.

Author Response

Hello,

Please find a file in the attachment containing detailed answers to your questions as well as revised manuscript as per your recommendations.

Sincerely,

Ali HUSSAN

Round 2

Reviewer 1 Report

Comments and Suggestions for Authors

After the needed corrections, now the manuscript is suitable to accept.

Reviewer 2 Report

Comments and Suggestions for Authors

The authors' revisions have sufficiently addressed the concerns, making their work suitable for recommendation for publication.